# Describing the sounds of nature: Using onomatopoeia to classify bird calls for citizen science

**Kellie Vella** ⬦ * ◉, **Daniel Johnson** ◉, **Paul Roe** ◉

Faculty of Science, Queensland University of Technology (QUT), Brisbane, Queensland, Australia

◉ These authors contributed equally to this work.
* kellie.vella@qut.edu.au

## Abstract

Bird call libraries are difficult to collect yet vital for bio-acoustics studies. A potential solution is citizen science labelling of calls. However, acoustic annotation techniques are still relatively undeveloped and in parallel, citizen science initiatives struggle with maintaining participant engagement, while increasing efficiency and accuracy. This study explores the use of an under-utilised and theoretically engaging and intuitive means of sound categorisation: onomatopoeia. To learn if onomatopoeia was a reliable means of categorisation, an online experiment was conducted. Participants sourced from Amazon mTurk (N = 104) ranked how well twelve onomatopoeic words described acoustic recordings of ten native Australian bird calls. Of the ten bird calls, repeated measures ANOVA revealed that five of these had single descriptors ranked significantly higher than all others, while the remaining calls had multiple descriptors that were rated significantly higher than others. Agreement as assessed by Kendall's *W* shows that overall, raters agreed regarding the suitability and unsuitability of the descriptors used across all bird calls. Further analysis of the spread of responses using frequency charts confirms this and indicates that agreement on which descriptors were unsuitable was pronounced throughout, and that stronger agreement of suitable singular descriptions was matched with greater rater confidence. This demonstrates that onomatopoeia may be reliably used to classify bird calls by non-expert listeners, adding to the suite of methods used in classification of biological sounds. Interface design implications for acoustic annotation are discussed.

## Introduction

Bio-acoustics is the study of animal calls to understand the distribution, behaviour and communication of different species [1]. Recently, call identification has been greatly aided by the development of automated call recognisers using machine learning. However, many forms of machine learning rely upon human intelligence to provide the pre-labelled datasets that they are trained upon, and the production of these annotated datasets is a time-consuming process [2]. While crowd-sourced human intelligence offers a potential solution, issues remain in terms of participant accuracy and efficiency [3, 4], as well as how to motivate continued

**Data Availability Statement:** All relevant data are within the paper and its Supporting Information files.

**Funding:** This study was supported by the Australian Research Council (www.arc.gov.au),

and awarded to PR (DP170104004). The funders had no role in study design, data collection and analysis, decision to publish, or preparation of the manuscript.

**Competing interests:** The authors have declared that no competing interests exist.

involvement in the task. However, citizen science offers additional benefits to ecological projects, such as engaging the public with scientific processes [5] and conservation agendas [6, 7]. As such, finding methods that both engage citizen scientists and allow the swift and accurate categorisation of complex vocalisations will provide an advantage over the output of a much smaller number of experts labouring alone, while also advancing ecological science in the public sphere.

Some of the method for acoustic annotation being explored include the pairing of short snippets of sound with visualisations such as spectrograms [3, 8], or just providing visualisations [9]. While this presents an advantage for representing certain species' calls (e.g. many bat calls are ultrasonic) and presents an additional layer of information upon which to make a judgement, they do not represent meaning intuitively. However, calls are also being identified using onomatopoeia [10], a method that may have a cognitive processing advantage [11–13]. For example, in 'Hawk Talk' [10], participants are tasked with watching video clips of a nesting hawk and categorising the vocalisations of the adults and chicks with terms such as 'peeping', 'chwirk', 'gank', and 'kee-eee-arrr'. The simplified options presented to citizen scientists in projects such as these stands in contrast to field observations in which the volume of data collected overwhelms analysis [14]. However, whether citizen scientists agree as to what constitutes a 'chwirk' is unknown and is the subject of this research. If proven so, the use of onomatopoeia may provide an effective means of communicating relationships between sounds and natural events that also provides a more engaging and streamlined citizen science experience.

The following describes an experimental study in which participants were tasked with identifying the most correct description of audio recordings of ten Australian native bird calls. We aimed to learn if onomatopoeia provides a reliable means of describing natural sounds among non-experts, to contribute to the development of new methods that aid bird call identification. This has implications for the annotation of bioacoustics call libraries, and applications within crowdsourced citizen science initiatives making use of audio datasets, such as the popular eBird [15]. We also discuss what the use of textual description as a classifying method might mean for the development of new interfaces.

## Citizens science and bioacoustics

Citizen science is recognised as having a strong contribution to make to environmental management and protection [16]. It has been described as resting upon two pillars: one concerned with the data management of distributed resources, and the other with initiating interest and maintaining motivation amongst volunteers [17]. As such, citizen science initiatives seek to balance what resources are devoted to developing techniques that capture large amounts of accurate data in the least amount of time, with those that provide an engaging experience for their participants. Citizen science focussed on ecological engagement also produces both direct and indirect benefits to both the participants and environmental conservation, by actively contributing to research agendas and fostering in the participant pro-environmental attitudes and behaviours [6, 7]. Additionally, further exploration of citizen science methods and interfaces may greatly aid bioacoustics research.

Within bioacoustics studies—the recording and analysis of acoustic recordings of vocal mammals, amphibians and insects [18]—birders have been particularly active, with the Xeno-Canto project highlighting the potential for sound bite sharing and classification amongst interested public. In contrast, the detection of calls within long-duration recordings require the aid of computer-assisted techniques that can quickly summarise and classify information, e.g. visualisation and call recognition algorithms. However, call recognisers require the

production of call libraries as training datasets, and call libraries have been found to be typically small in size, with low variability in the call types that are identified [2]. The pipeline for citizen science contributions to these libraries is being rapidly developed, with some models calling for ways to balance the input of experts, the crowd, and machine learning [19, 20], and forms of machine learning that reduce the amount of annotation needed [20]. One identified weakness in the pipeline is the need for acoustic annotation interfaces designed with human-use in mind [20].

Another challenge in the production of annotated datasets is the accuracy of data classification when performed by non-experts. Promising citizen science work has been carried out in projects utilising still images, such as *Snapshot Serengeti*, which captured data from 225 camera traps set out over the Serengeti National Park and crowd-sourced their categorisation with 96.6% accuracy [21]. However, fewer projects utilise participants to categorise audio, perhaps due to the added challenge of working with temporally changing data [4]. One project which did attempt this, carefully curated the audio presented to participants. *Bat Detective*, presented citizen scientists with 3.84 second long sound clips, utilising time-expanded audio (allowing bat calls to be easily heard), and an accompanying visualisation [8]. Annotation was carried out on these visualisations utilising three examples of search-phase echo-location calls, terminal feeding buzzes, and social calls. However, due to many observed errors, instead of merging annotations, they chose to use a subset of their most prolific and accurate annotator's work to train a convolutional neural network. This issue of annotator accuracy and efficiency is particularly pertinent to audio annotation with best practice still being established.

Task complexity also plays a part in shaping audio annotation accuracy and efficiency, as does the interplay between complexity and participant type. A study considering differences between paid crowdworkers and citizen scientists found that while paid crowdworkers preferred hierarchical binary-labelling tasks (that can be completed quickly), citizen scientists were more comfortable with single pass multi-labelling tasks [4]. This may be because the greater complexity of the multi-labelling task acted as a learning activity for already intrinsically motivated participants. In turn, this suggests a place for more complex data when engaging citizen scientists, particularly if utilising methods that facilitate intuitive recall.

The usefulness of task complexity is supported by research finding that citizen scientists benefit from a media assortment when exploring acoustic data [22]. This study provided a group of participants with media related to three species of Australian birds: spectrograms, photographs, audio recordings, and distribution maps. These media were presented in that order to all participants until all media were present, and their conversational sense-making analysed. Having access to a range of media allowed participants to cross match and resolve ambiguity and was enthusiastically received. In this group setting, the variety of media also provided opportunities for discussing calls, which in turn led to the spontaneous recreation and simplification of sounds, e.g. "'*Raak, raak*', *very clearly to me spells out that call*" (p.1693). This suggests that text descriptions could be another useful addition to the media set by making links between sounds and images through the reiteration of patterns (e.g. repeating the same sound), or further describing the qualities of the sound (e.g. lower frequencies). The value of combining different types of information is also supported by a study that contrasted the labelling of audio clips using semantic descriptions and onomatopoeia [23]. The model combining both approaches was found to produce most accurate labelling when applied to a mixture of natural and human-made sounds. The assemblage of different ways of reporting the same event may be even more important for the identification of sounds in online settings without the recourse of collaborative discussion. Considering onomatopoeia as an additional classification method may also offer other benefits, such as being a more intuitive means of understanding natural sounds.

## Onomatopoeia and meaning

Onomatopoeic words are a type of sound symbolism, in which nouns or verbs that are similar to the sound that a thing or action makes are used to describe it, for example, 'sizzle' or 'screech'. They are present across a wide range of languages and are particularly present in the descriptions of the sounds of nature (for a list of animal sounds across 17 languages: http://www.eleceng.adelaide.edu.au/Personal/dabbott/animal.html). The use of sound-symbolic words is also very apparent in bird naming, with one ethnographic study finding that more than a third of the terms for bird species were an onomatopoeic reference to the calls the birds made [24]. Amusing variations may also assist in the memorising of call patterns, e.g. the White-throated Sparrow's call is variably reported as 'Oh-sweet-Canada-Canada' or 'Old-Sam-Peabody-Peabody' [25]. Additionally, current research demonstrates advantages for onomatopoeic words in early perception, production and interaction [11–13, 26–30]. This may be due to their simplistic phonological nature and phonetic flexibility, making them easier for infants to acquire and produce, as well as the iconic nature that allows them to be understood more easily.

Studies of sound symbolism establish its cognitive advantage in terms of tying sound to meaning in early life. Imai and Kita [27] outline the evidence for the sound symbolism boot-strapping hypothesis for language evolution. The authors suggest that sound symbolism helps infants gain references for speech sounds, establish a lexical representation, and to learn words by allowing them to focus on referents in a complex scene. This roadmap supports the early acquisition of sound-symbolic words, suggesting that onomatopoeic sound descriptions may be easier to discriminate due to an early processing advantage, and is supported in studies utilising Japanese [28], and English [29, 30] words. Onomatopoeic words also appear to provide an advantage for adult retrieval of meaning [11–13], which has also been shown to be the case across both non brain damaged and aphasic speakers [12]. This highlights a processing advantage for words acquired early in life, which could explain why onomatopoeic sound-words, that are thought to be acquired early in life, are discriminated easier than non-onomatopoeic words [12]. However, sound symbolism has also been applied to the adult acquisition of new languages, as is demonstrated in a study by Lockwood, Dingemanse and Hagoort [31]. In this, Dutch-speaking adults were found to learn Japanese words better when they were ideophonic: sound-symbolic words that depict diverse sensory information, e.g. sound, but also taste, visual effects, etc. This suggests that sound-symbolism is not only important for language development in childhood, but also aids the linking of sound to meaning in adulthood, and across languages.

Our intuitive understanding of onomatopoeia may be due to it falling on the showing-saying continuum [32]—in that onomatopoeic words are used in communication as an attempt to recreate the sensory experience by using sounds that provide an accurate representation of the experience. Similarly, sounds that can be embodied may be easier to recognise as there is an automatic processing advantage to "embody the sound if possible" [33]. This may explain why sounds that are described by onomatopoeic words that are a direct representation of their true sound, are easier to discriminate from other sound description words. While few studies investigate the relationship between meaning and sound processing using animal sounds, one experiment utilising Japanese-speaking adults suggests that animal-referring onomatopoeia acts as a bridge between nouns and animal sounds [34]. In this study, participants completed a sound categorisation task in which they heard four types of sound stimuli: onomatopoeic sounds (verbally spoken words), animal sounds (non-verbal), nouns (verbally spoken), and a pure tone/noise (control) and were tasked with identifying which sounds related to birds, and which did not. Each sound category was found to be associated with different brain regions,

with onomatopoeias activating extensive brain regions that are involved in the processing of verbal and non-verbal (animal) sounds. However, behavioural data demonstrated no significant difference in accuracy or reaction times between sound categories. As this study made use of the Japanese language, further research is required to learn if onomatopoeia might be a useful technique to enable English-speaking participants to quickly categorise animal sounds.

Sound embodiment also suggests that onomatopoeia can be used to communicate combinations of sound qualities such as pitch, duration, and oscillation. Initial support for this idea is found in a study experimentally testing similarities between onomatopoeia and their corresponding sounds, and finding that duration and frequency of sounds corresponded to both the vowels and consonants of uttered onomatopoeia [35]. Anecdotal evidence from within the hearing-impaired community also suggests it is so, with one teacher describing how ten-year old hearing-impaired students comprehended an onomatopoeic word: "*They could not hear the actual sound "ping." But they understood immediately when I pinged a rubber-band in sign language*" [36]. This suggests that short duration and sharpness of the motion of flicking a rubber-band communicated the qualities of the sound. If onomatopoeia does offer a means to quickly summarise a range of sound qualities, it potentially provides an intuitive means of quickly categorising the rich and variable natural sounds within citizen science projects.

In summary, onomatopoeia offers advantages for sound summarisation and recall that may aid citizen science focused on natural sounds. While current bioacoustics research already makes use of visual and auditory aids to enable the annotation of acoustic datasets, the addition of text-based categories may offset error and increase engagement. In turn, this will support the development of training datasets on which to test bioacoustics machine-learning tools and techniques. However, whether non-experts can agree on onomatopoeic descriptors to categorise natural sounds is unknown. Clarifying this is an important first step in the development of text-based categorisation tools for bioacoustics research.

## Method and materials

### Procedure

All participants indicated their consent to take part in the study electronically before participating. Ethical approval was gained from Queensland University of Technology's ethics board to conduct low-risk human research (project number: 1700001062).

Participants were guided to a website with embedded sound files, which provided the calls of ten different Australian native birds in counter-balanced order (see Table 1 for a list of species and a link to their sound file). After each sound file they were presented with a range of call descriptions, namely: 'chatter', 'trill', 'warble', 'whistle', 'moan', 'chirp', 'hoot', 'click', 'buzz', 'cackle', 'screech', and 'peep'. These descriptions were chosen by two of the authors after reviewing the literature and reducing the possible options to those that captured the largest range of commonly used bird call descriptors. Amateur field guides were consulted, as well as descriptors used by Birdlife Australia's *Birds in Backyards* (a research, education and conservation program mapping Australian birds). We did not seek to capture functional classes of sounds (e.g. alarm calls, or songs), but instead sought sound descriptors that were commonly applied to a broad range of bird calls. Kookaburra calls and the descriptor 'laugh' were excluded due to the strong Australian cultural association between the words 'kookaburra' and 'laugh'. Participants were asked to rate each of these descriptors in terms of how well each word described the sound, using a five-point Likert scale from 1 'Not at all' to 5 'Extremely well'. This approach was favoured over the use of open-ended questions, i.e. asking participants to describe the calls, as we wished to first establish whether descriptors used within

**Table 1. Represented Australian native bird species, links to their recorded calls, and average number of times these calls were listened to.**

| Common name | Recording link | Mean number of times sound file played (SD) |
|---|---|---|
| *scientific name* | | |
| Channel Billed Cuckoo | https://youtu.be/nVtr_dwqQsU | 1.73 (1.72) |
| *Scythrops novaehollandiae* | | |
| Southern Boobook | https://youtu.be/UcqZ_L-doLA | 1.58 (1.67) |
| *Ninox novaeseelandiae* | | |
| Tawny Frogmouth | https://youtu.be/9uLnXl1EuSU | 1.71 (2.1) |
| *Podargus strigoides* | | |
| Masked Lapwing | https://youtu.be/uhTDZdayyvI | 1.8 (2.1) |
| *Vanellus miles* | | |
| Little Wattlebird | https://youtu.be/HQkSEu8sC8I | 1.61 (1.23) |
| *Anthochaera chrysoptera* | | |
| Magpie Lark | https://youtu.be/U3wYrGHf1ZM | 1.72 (1.99) |
| *Grallina cyanoleuca* | | |
| Black Faced Cuckoo Shrike | https://youtu.be/DI3jMFsTeIo | 1.51 (1) |
| *Coracina novaehollandiae* | | |
| Satin Bowerbird | https://youtu.be/Iw2l0AqOqRQ | 1.57 (1.3) |
| *Ptilonorhynchus violaceus* | | |
| Green Catbird | https://youtu.be/habq9W3cYw8 | 1.7 (1.95) |
| *Ailuroedus crassirostris* | | |
| Bush Stone Curlew | https://youtu.be/27HH1nK2ktw | 1.63 (2.26) |
| *Burhinus grallarius* | | |

birding communities would facilitate agreement, and an open-ended question was likely to produce a mixture of these as well as invented descriptions.

After rating each call, the participants were asked to approximate how many times they played the sound (averages provided in Table 1). Participants were also asked to rate how confident they felt about their ratings ('confidence ratings') of each bird call using a five-point Likert scale from 1 'Not at all' to 5 'Extremely well', which were then averaged. The entire task took 9.68 minutes (SD = 3.98 minutes) to complete, on average.

## Participants

Participants were sourced from Amazon Mechanical Turk (mTurk)—an online marketplace brokering human intelligence tasks. An initial pool of 115 participants received US$1.40 compensation for their time. mTurk workers' age and gender were not taken at the time of the survey, however, mTurk demographic data (as of August 7, 2018) shows them to be 53.19% female, and the majority (78.72%) born between 1990 and 2000. Only mTurk workers with qualification of being 'Masters' were engaged (indicating that they were highly-rated workers with a good grasp of the English language). Prior to analysis participants that did not rate all bird sounds in the task, and took less than five minutes to complete the task were excluded. This resulted in 11 participants being removed, leaving 104 participants for analysis.

## Analysis

Multiple one-way repeated measures ANOVA (RM ANOVA) were conducted using *IBM SPSS Statistics* (2017). These analyses provide an indication of whether particular words were rated as being significantly more or less applicable to each sound recording. In all cases, Mauchly's test of the assumption of sphericity was breached for each test, and a Greenhouse-

Geisser estimate of sphericity was used to correct degrees of freedom [37], with statistics listed. However, all other statistical assumptions were met. The threshold for statistical significance was set at $p < .05$. As there were many planned comparisons, only Bonferroni-corrected significant comparisons with an effect size greater than $d = .80$ were determined as statistically relevant for discussion.

Additionally, the overall levels of agreement regarding the ratings given to each bird call's set of descriptions, were tested using Kendall's $W$. All assumptions were met. To facilitate understanding of differences in agreement between individual descriptions, these statistical analyses are supplemented with a graphical depiction of the frequency with which each numbered scale point was selected by participants for each recording, combined with self-reported confidence ratings.

All mean congruence ratings, descriptive statistics, and comparisons showing mean differences, $p$ values and effect sizes are provided as supplementary material, as is a one-page collection of all frequency charts for easy comparison.

## Results

### Channel Billed Cuckoo analysis

Mauchly's test revealed the assumption of sphericity was breached for the main effect of bird sound description ($\chi^2(65) = 325.52$, p < .001), and a Greenhouse-Geisser estimate of sphericity ($< .75$), was used to correct degrees of freedom ($\varepsilon = .62$). Analysis revealed a significant main effect of bird sound description ($F (6.859, 631.02) = 39.91$, p < .001, $\eta_p^2 = .30$). On average, participants rated the sound description "screech" with the highest rating, and "buzz" with the lowest rating. Bonferroni corrected post-hoc mean difference comparisons revealed "screech" was rated significantly higher than all other bird sound descriptors (all p's < .001, all d's > 1.05). Additionally, "buzz" was rated significantly lower than "trill" (mean difference = -.86, p < .001, d = -.82, 95% CI [-1.35, -.37]). Participants significantly agreed in their rating of the supplied descriptors for the call of the Channel Billed Cuckoo, $W = .231$, $p < .001$. See Fig 1.

### Southern Boobook analysis

Mauchly's test revealed the assumption of sphericity was breached for the main effect of bird sound description ($\chi^2(65) = 322.30$, $p < .001$), and a Greenhouse-Geisser estimate of sphericity ($< .75$), was used to correct degrees of freedom ($\varepsilon = .54$). Analysis revealed a significant main effect of bird sound description ($F (5.93, 545.15) = 62.52$, $p < .001$, $\eta_p^2 = .41$). On average, participants rated the sound description "hoot" with the highest rating, and "click" with the lowest rating. Bonferroni corrected post-hoc mean difference comparisons revealed "hoot" was rated significantly higher than all other bird sounds (all $p$'s < .001, all $d$'s > 1.26). Participants significantly agreed in their rating of the supplied descriptors for the call of the Southern Boobook, $W = .296$, $p < .001$. See Fig 2.

### Tawny Frogmouth analysis

Mauchly's test revealed the assumption of sphericity was breached for the main effect of bird sound description ($\chi^2(65) = 469.72$, $p < .001$), and a Greenhouse-Geisser estimate of sphericity ($< .75$), was used to correct degrees of freedom ($\varepsilon = .48$). Analysis revealed a significant main effect of bird sound description ($F (5.25, 483.10) = 98.82$, $p < .001$, $\eta_p^2 = .52$). On average, participants rated the sound description "hoot" with the highest rating, and "click", and "cackle" with equal lowest ratings. Bonferroni corrected post-hoc mean difference comparisons revealed "hoot" was rated significantly higher than all other bird sounds (all $p$'s < .001, all

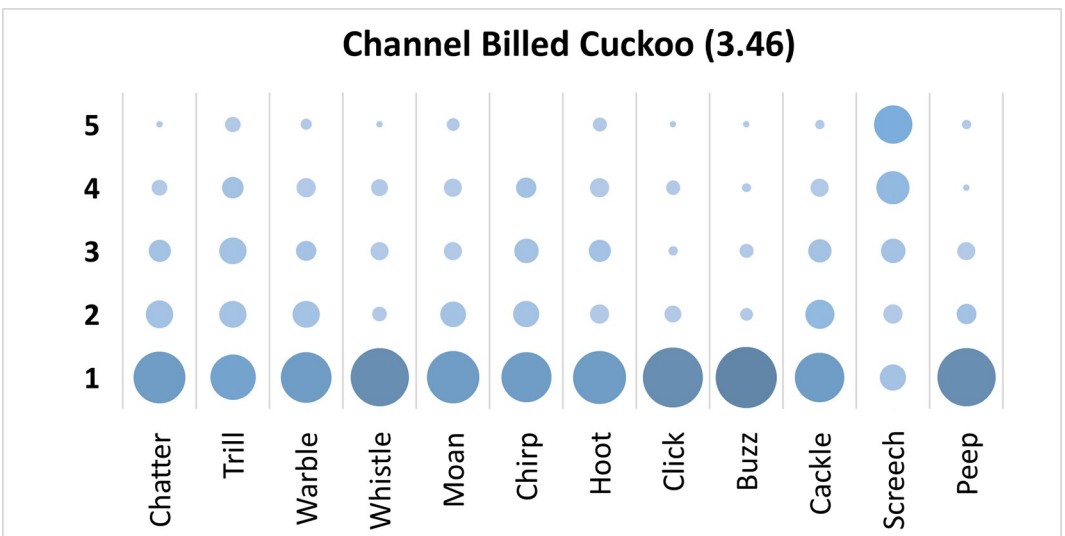

**Fig 1. Count of participants' sound descriptions scores for the Channel Billed Cuckoo with confidence ratings in brackets.**

d's > 1.69). Additionally, participants rated "moan" significantly higher than "cackle" (mean difference = .74, p < .001, d = .82, 95% CI [.36, 1.13]), and "click" (mean difference = .74, p < .001, d = .81, 95% CI [.34, 1.15]).

Participants significantly agreed in their rating of the supplied descriptors for the call of the Tawny Frogmouth, $W$ = .387, $p$ < .001. See Fig 3.

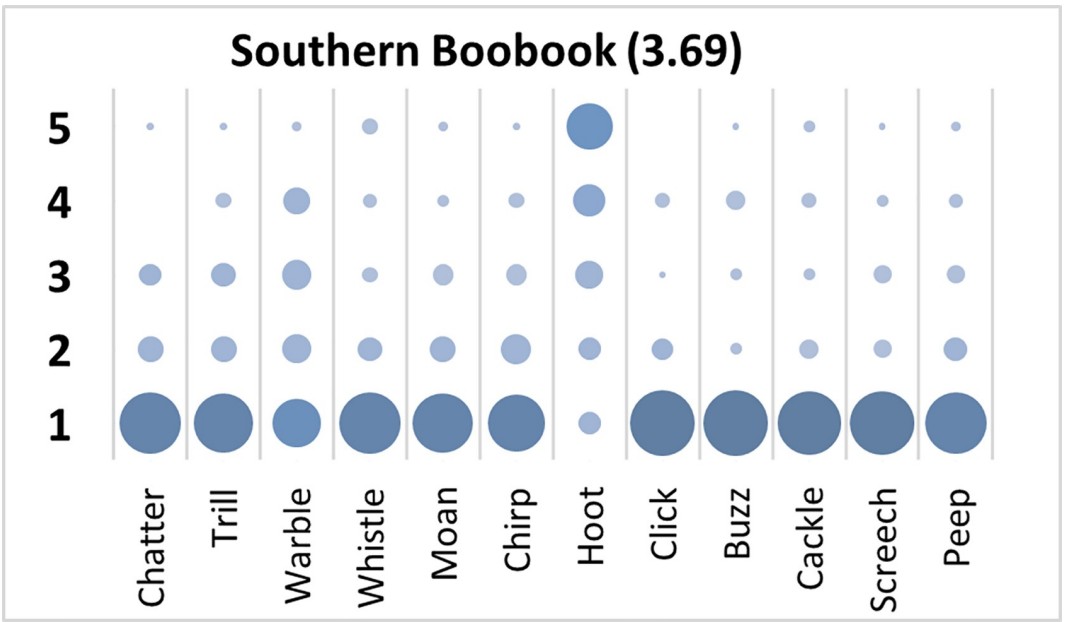

**Fig 2. Count of participants' sound descriptions scores for the Southern Boobook call with confidence ratings in brackets.**

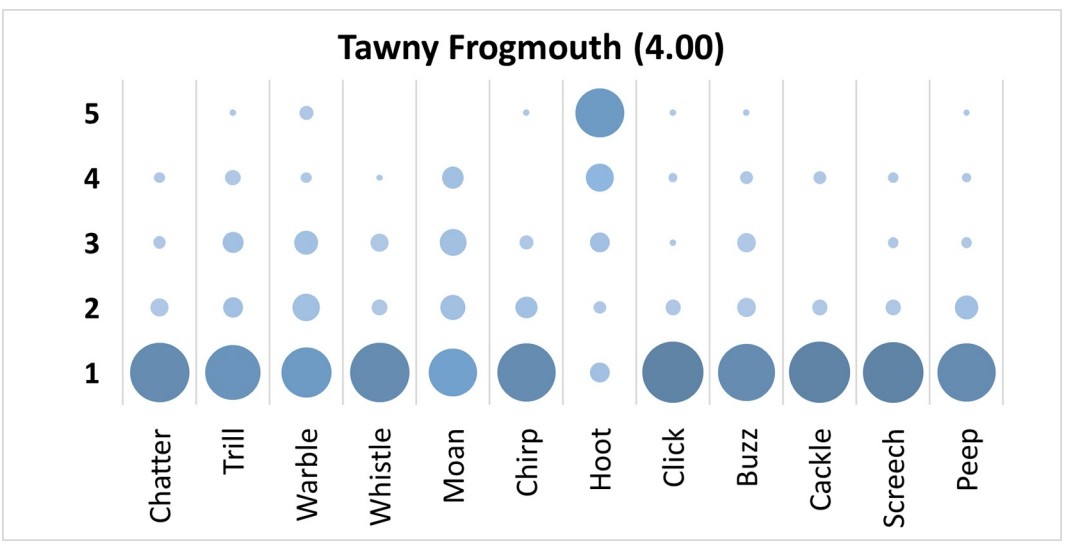

**Fig 3. Count of participants' sound descriptions scores for the Tawny Frogmouth with confidence ratings in brackets.**

## Masked Lapwing analysis

Mauchly's test revealed the assumption of sphericity was breached for the main effect of bird sound description ($\chi^2(65) = 284.01$, $p < .001$), and a Greenhouse-Geisser estimate of sphericity ($< .75$), was used to correct degrees of freedom ($\varepsilon = .64$). Analysis revealed a significant main effect of bird sound description ($F(7.03, 646.50) = 38.43$, $p < .001$, $\eta_p^2 = .30$). On average, participants rated the sound description "chatter" with the highest rating, and "moan" with the lowest rating. Bonferroni corrected post-hoc mean difference comparisons revealed no one clearly preferred description. "Chatter" was rated significantly higher than "moan", "hoot", "buzz", "whistle", "peep", "click", and "warble" (all $p$'s $< .001$, all $d$'s $> 1.08$). Additionally, "cackle" and "chirp" were both rated significantly higher than "moan", "hoot", "buzz", "whistle", "peep", and "click" (all $p$'s $< .001$, all $d$'s $> .81$). Finally, "screech" and "trill" were both rated significantly higher than "moan", "hoot", "whistle", and "buzz" (all $p$'s $< .001$, all $d$'s $> .84$). Participants significantly agreed in their rating of the supplied descriptors for the call of the Masked Lapwing, $W = .296$, $p < .001$. See Fig 4.

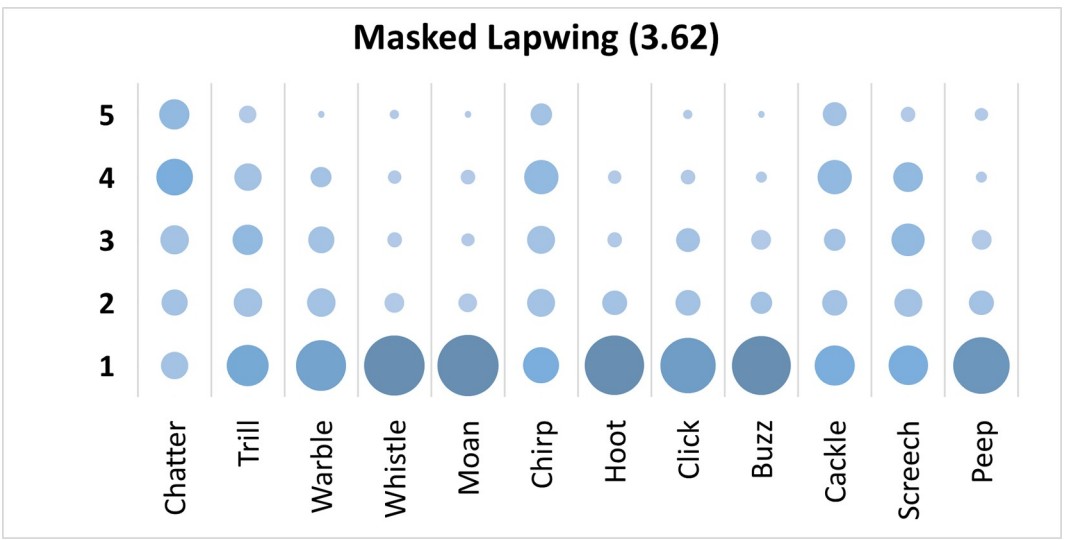

**Fig 4. Count of participants' sound descriptions scores for the Masked Lapwing with confidence ratings in brackets.**

## Little Wattlebird analysis

Mauchly's test revealed the assumption of sphericity was breached for the main effect of bird sound description ($\chi^2(65) = 285.79$, p < .001), and a Greenhouse-Geisser estimate of sphericity (< .75), was used to correct degrees of freedom ($\varepsilon = .69$). Analysis revealed a significant main effect of bird sound description ($F$ (7.62, 701.01) = 30.84 p < .001, $\eta_p^2 = .25$). Again, there was no one clearly preferred description. On average, participants rated the sound description "chatter" with the highest rating, and "buzz" with the lowest rating. Bonferroni corrected post-hoc mean difference comparisons revealed "chatter" was rated significantly higher than "buzz", "moan", "hoot", "click", "whistle", "peep", and "trill" (all p's < .001, all d's > .84). Additionally, "Chirp" was rated significantly higher than "buzz", "moan", "hoot", "click", "whistle" and "peep" (all p's < .001 all d's > .80); and "Cackle" was rated significantly higher than "hoot", "moan", and "buzz" (all p's < .001 all d's > .94). Finally, "screech" was rated significantly higher than "buzz" (mean difference = .94, p < .001, d = .88, 95% CI [.49, 1.38]). Participants significantly agreed in their rating of the descriptors for the call of the Little Wattlebird, $W = .255$, $p < .001$. See Fig 5.

## Magpie Lark analysis

Mauchly's test revealed the assumption of sphericity was breached for the main effect of bird sound description ($\chi^2(65) = 290.5$, p < .001), and a Greenhouse-Geisser estimate of sphericity (< .75), was used to correct degrees of freedom ($\varepsilon = .61$). Analysis revealed a significant main effect of bird sound description ($F$ (6.76, 621.49) = 36.15, p < .001, $\eta_p^2 = .28$). On average, participants rated the sound description "chirp" with the highest rating, and "moan" with the lowest rating. Bonferroni corrected post-hoc mean difference comparisons revealed "chirp" was rated significantly higher than "moan" "hoot", "buzz", "click", "warble", "cackle", "whistle", and "trill" (all p's < .001, all d's > .90). However, "chatter", "screech", and "peep" were also rated significantly higher than "hoot", "click", "moan", and "buzz" (all p's < .001 all d's > .90). Participants significantly agreed in their rating of the supplied descriptors for the call of the Magpie Lark, $W = .286$, $p < .001$. See Fig 6.

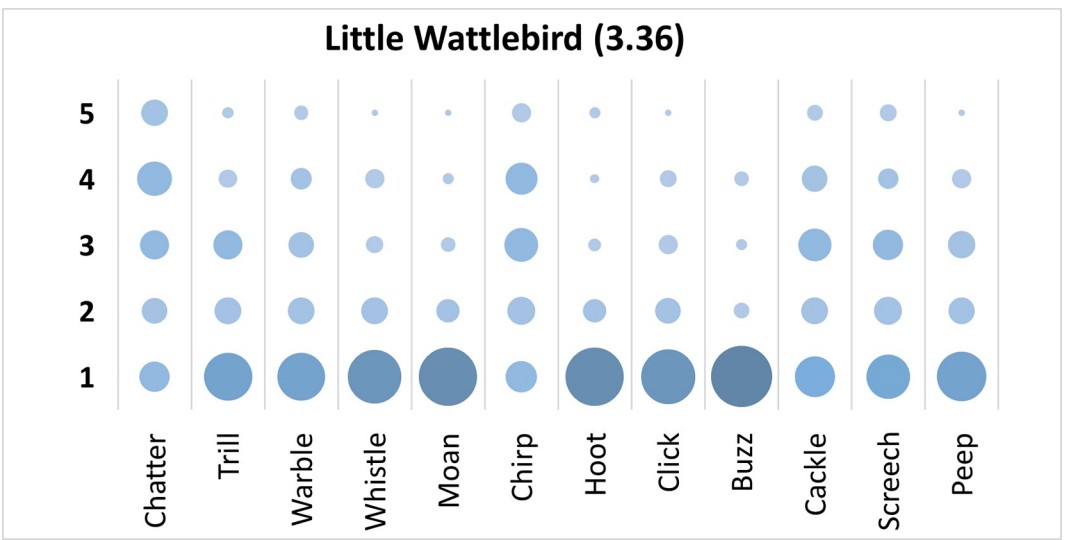

**Fig 5. Count of participants' sound descriptions scores for the Little Wattlebird with confidence ratings in brackets.**

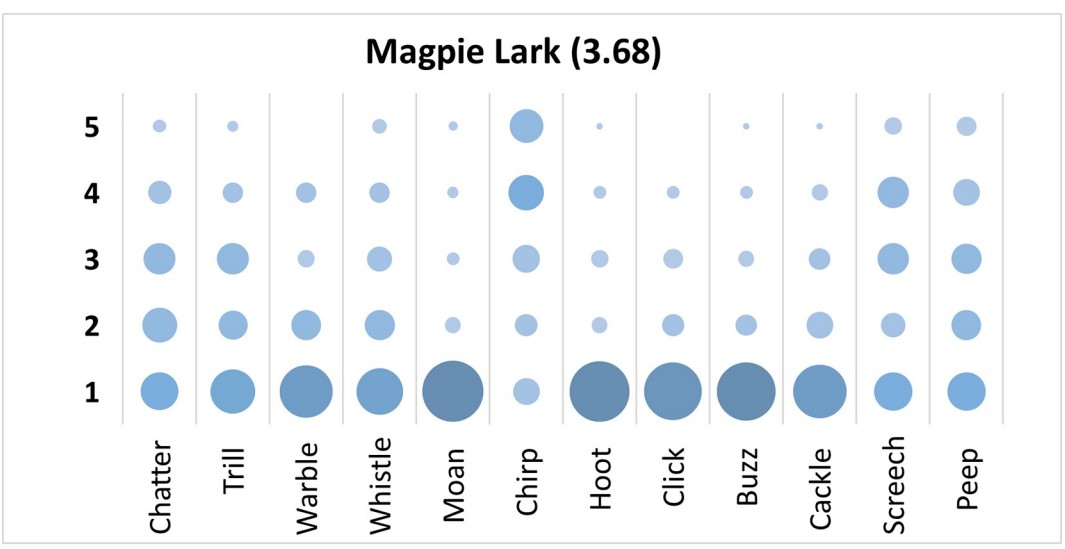

**Fig 6. Count of participants' sound descriptions scores for the Magpie Lark with confidence ratings in brackets.**

### Black Faced Cuckoo Shrike analysis

Mauchly's test revealed the assumption of sphericity was breached for the main effect of bird sound description ($\chi^2(65) = 351.28$, p < .001), and a Greenhouse-Geisser estimate of sphericity (< .75), was used to correct degrees of freedom ($\varepsilon = .63$). Analysis revealed a significant main effect of bird sound description ($F(6.96, 640.72) = 41.70$, p < .001, $\eta_p^2 = .31$). On average, participants rated the sound description "chirp" with the highest rating, and "moan" with the lowest rating. Bonferroni corrected post-hoc mean difference comparisons revealed "chirp" was rated significantly higher than all other bird sounds (all p's < .001, all d's > .93). Additionally, "trill" was rated significantly higher than "moan", "hoot", and "click" (p < .001, d = .83). Participants significantly agreed in their rating of the supplied descriptors for the call of the Black Faced Cuckoo Shrike, $W = .272$, $p < .001$. See Fig 7.

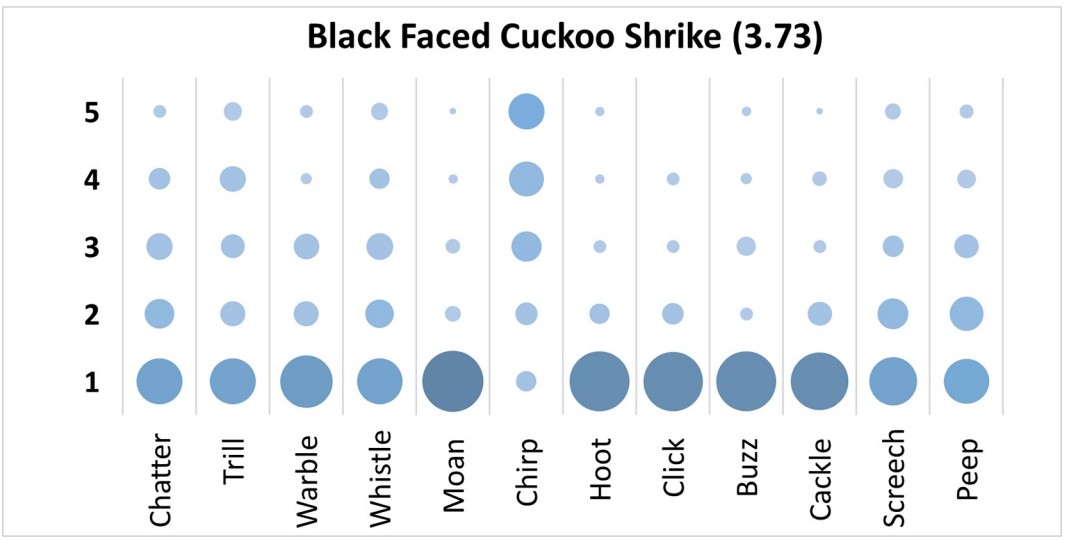

**Fig 7. Count of participants' sound descriptions scores for the Black Faced Cuckoo Shrike with confidence ratings in brackets.**

### Satin Bowerbird analysis

Mauchly's test revealed the assumption of sphericity was breached for the main effect of bird sound description ($\chi^2(65) = 433.54$, p < .001), and a Greenhouse-Geisser estimate of sphericity (< .75), was used to correct degrees of freedom ($\varepsilon = .49$). Analysis revealed a significant main effect of bird sound description ($F (5.37, 493.89) = 33.88$, p < .001, $\eta_p^2 = .27$). On average, participants rated the sound description "screech" with the highest rating, closely followed by "buzz", while "hoot" was rated as the least adequate sound description. Bonferroni corrected post-hoc mean difference comparisons revealed "screech", and "buzz" were both rated significantly higher than "hoot", "whistle", "peep", "warble", "chirp", "click", "chatter", and "moan" (all p's < .001, all d's > .96). Participants significantly agreed in their rating of the supplied descriptors for the call of the Satin Bowerbird, $W = .227$, $p < .001$. See Fig 8.

### Green Catbird analysis

Mauchly's test revealed the assumption of sphericity was breached for the main effect of bird sound description ($\chi^2(65) = 459.02$, p < .001), and a Greenhouse-Geisser estimate of sphericity (< .75), was used to correct degrees of freedom ($\varepsilon = .56$). Analysis revealed a significant main effect of bird sound description ($F (6.11, 562.44) = 46.49$, p < .001, $\eta_p^2 = .34$). On average, participants rated the sound description "screech" highest, and "click" with the lowest rating. Bonferroni corrected post-hoc mean difference revealed "screech" was rated significantly higher than "click", "whistle", "hoot", "peep", "chirp", "buzz", "trill", "warble", "chatter", and "cackle" (all p's < .001, all d's > .94). "Moan" was rated significantly higher than "click", "whistle", "hoot", "peep", "chirp", "warble", "buzz", "chatter", and "trill" (all p's < .001, all d's > .96). Finally, "cackle" was rated significantly higher than "click" (mean difference = .80, p < .001, d = .84, 95% CI [.39, 1.20]). Participants significantly agreed in their rating of the supplied descriptors for the call of the Green Catbird, $W = .295$, $p < .001$. See Fig 9.

### Bush Stone Curlew analysis

Mauchly's test revealed the assumption of sphericity was breached for the main effect of bird sound description ($\chi^2(65) = 252.78$, p < .001), and a Greenhouse-Geisser estimate of sphericity

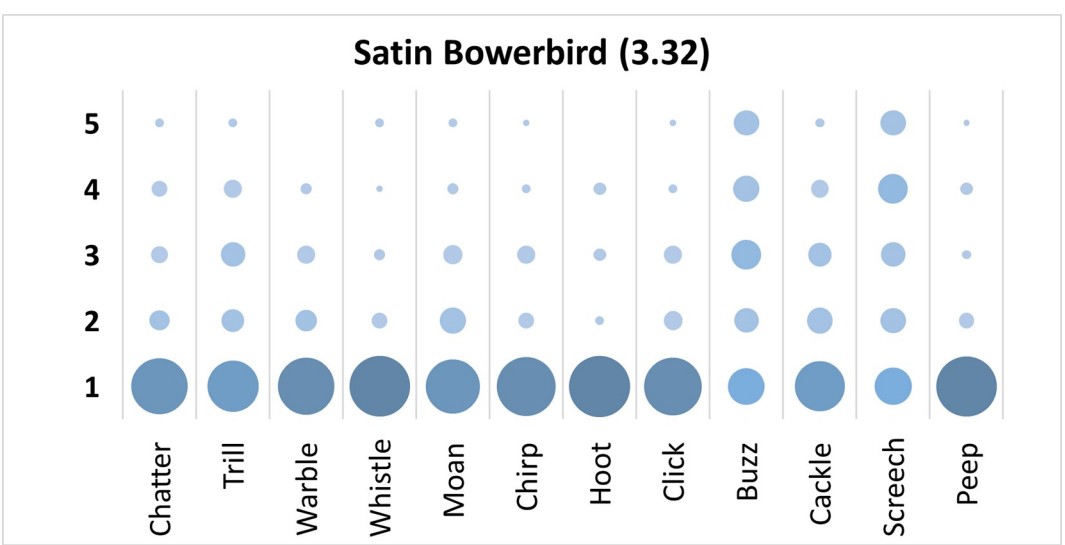

**Fig 8. Count of participants' sound descriptions scores for the Satin Bowerbird with confidence ratings in brackets.**

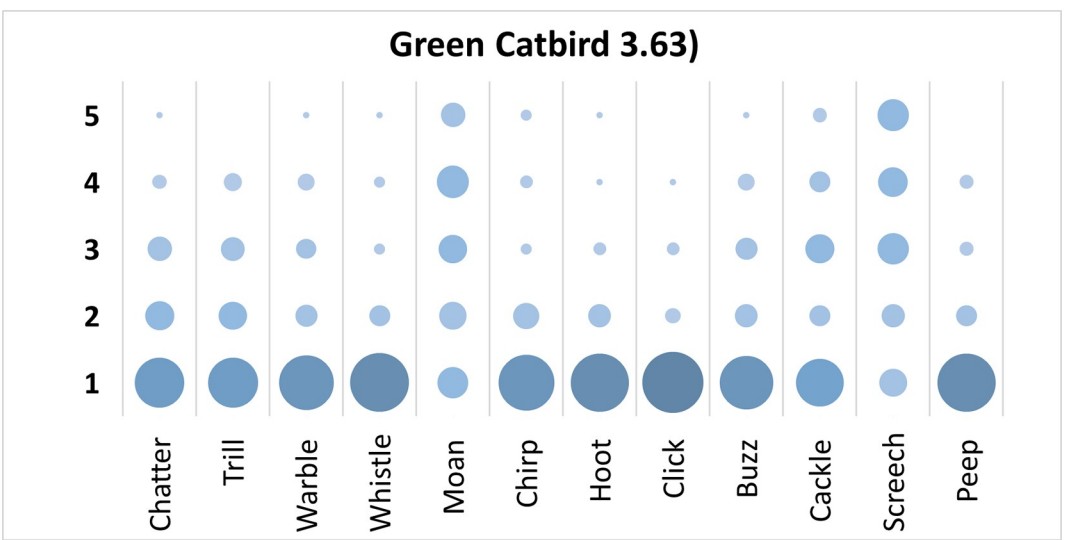

**Fig 9. Count of participants' sound descriptions scores for the Green Catbird with confidence ratings in brackets.**

($<$ .75), was used to correct degrees of freedom ($\varepsilon$ = .68). Analysis revealed a significant main effect of bird sound description ($F$ (7.52, 691.41) = 63.75, p $<$ .001, $\eta_p^2$ = .41). On average, participants rated the sound description "whistle" with the highest rating, and "click" with the lowest rating. Bonferroni corrected post-hoc mean difference comparisons were analysed revealed "whistle" was rated significantly higher than all other bird sounds (all p's $<$ .001, all d's $>$ .1.64). Additionally, "click" was rated significantly lower than "chirp", (Mean difference = -.88, p $<$ .001, d = -.82, 95% CI [-1.32, -.44]). Participants significantly agreed in their rating of the supplied descriptors for the call of the Bush Stone Curlew, $W$ = .307, $p <$ .001. See Fig 10.

## Discussion

This study establishes that certain descriptors were rated more highly than others, and that raters agreed regarding the suitability and unsuitability of various descriptors, for all bird calls.

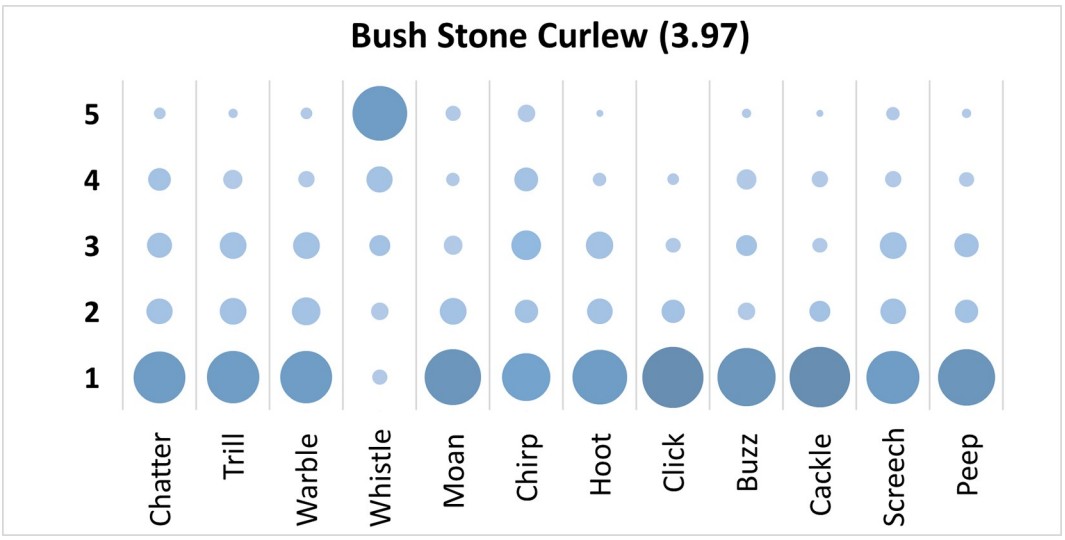

**Fig 10. Count of participants' sound descriptions scores for the Bush Stone Curlew with confidence ratings in brackets.**

There were also variations in the level of agreement as assessed by Kendall's *W*. Visual inspection of frequency charts showed mixed agreement regarding the suitability of different descriptors, and strong agreement regarding the unsuitability of many descriptors throughout. While only half of the bird calls could be adequately described using one of the given descriptors, the evidence shows that people largely tend to agree on the best and worst descriptors for bird sounds—a necessary precondition for the use of onomatopoeia in citizen science. Put another way, if there was very little agreement among people on how to describe or not describe specific bird sounds this would indicate that experts were needed to effectively categorise bird sounds. Instead, these findings support onomatopoeia being added to the suite of methods that can be used in acoustic categorisation tasks and confirms the value of further exploring its use in citizen science annotated bioacoustics call libraries.

The presence of multiple significant differences and lower levels of agreement for some bird calls, however, suggests the necessity of carefully considering the choice of descriptors. It is possible that with a larger set of either calls or descriptors, a greater number of strongly agreed-upon, highly rated descriptors would have presented (e.g. 'hoot' for the Tawny Frogmouth, or 'whistle' for the Bush Stone Curlew). The finding that some bird calls had multiple descriptors being more highly rated than others may also be explained by the complexity of some bird calls. For example, both 'buzz' and 'screech' were rated highly for the Satin Bowerbird, while the Green Catbird call drew the highest ratings from 'screech' and 'moan'. The vocal uniqueness of some species' calls might make concise textual summation difficult using singular, common onomatopoeic words. This argues for the use of onomatopoeia in combination with other forms of data, e.g. as part of a larger classification method that incorporates other features such as location and time of the call, or the use of different media (e.g. visualisation). Currently, visual and aural information is being used to guide categorisation tasks with mixed results when engaging citizen scientists [4, 8]. Other research has found that an assortment of media can resolve ambiguity when trying to make sense of ecological data that includes acoustics [22]. The addition of textual descriptors (onomatopoeia) to the mix should provide richer information on which citizen scientists can base their judgement in acoustic categorisation tasks, and cross-check categorisations.

Given our use of Australian native birds, which would sound unfamiliar to many listeners (even Australians), our findings suggest real value in exploring the use of onomatopoeia in natural sound categorisation. Onomatopoeia not only links sound to meaning in early life [11–13, 26–30], but as this study demonstrates, can also in adulthood be intuitively understood as a summarisation of a range of acoustic qualities, such as duration, pitch and oscillation. This is supported by research finding that the frequency and duration of sounds are tied to onomatopoeic meaning [35], suggesting that sound qualities are the foundation of new onomatopoeia. In doing so, onomatopoeia could contribute to reducing a gap within active machine learning models, by providing an engaging and intuitive human intelligence categorisation strategy [20], in a model making use of novices, experts and machine-learning [19]. If so, its facilitation of citizen science-led categorisation overcomes some of the practical and time-consuming steps in call categorisation, which hamper the production of a greater number of verified call libraries, containing varied call types [2]. Additionally, onomatopoeia may be an inherently engaging and intuitive means to label sounds, supported by research demonstrating its mental processing advantages [11–13], and its use as an adult language learning aid [31]. This, in turn, has ramifications for expediting acoustic annotation using crowd-sourced human intelligence, particularly when supported by expert oversight [20]. Though annotator accuracy using this method in comparison to others is yet to be explored, this study suggests that facilitating categorisation through the development of new interfaces that account for textual descriptions is warranted.

Patterns of ratings, as indicated by RM ANOVA across the ten native Australian bird calls, indicate opportunities in designing citizen science interfaces for acoustic annotation. The presence of repeat opposing high/low ratings suggest that certain sound descriptors represent contrasting sound properties. For example, repeat opposing high/low ratings for 'chatter' and 'buzz' (Little Wattlebird, Satin Bowerbird), or 'hoot' and 'click' (Southern Boobook, Tawny Frogmouth), suggest that these sound descriptors differ across multiple acoustic dimensions, such as duration, pitch, or oscillation. This understanding could be applied in user interfaces that allow citizen scientists to quickly sort the most appropriate descriptors (whether one or many) into clusters or rate sounds by using a slider with either sound presented as a diametrically opposing pair (e.g. 'this call is more like a click than a hoot'). Another possible solution is the invention of tailored onomatopoeic words that approximate bird calls, such as are being utilised by Cornell's Hawk Talk [10]. In both cases, complex qualities can potentially be reduced to simple and engaging choices, which could greatly improve the user experience of acoustic annotation and address the challenge of maintaining participant interest in the task [17].

Overall, our findings suggest that onomatopoeia is a viable categorisation method for some natural sounds, with the proviso that the suitability of descriptors needs to be determined in advance. We also make the case for exploring interfaces that enable complex sound categorisation to enhance the engagement and accuracy of citizen science audio annotation tasks and make suggestions for future research.

## Next steps

The usefulness of text descriptions may well be limited to vocal species with distinctly different calls that lend themselves to onomatopoeic representation, as the categorisation of less audibly distinct calls (e.g. bats) may produce a greater number of errors even when accompanied by visualisations [8]. Future research could engage with a greater number of species and types of calls to establish the limits of onomatopoeic descriptions. Similarly, confirmation that a greater number of onomatopoeic descriptions can be consistently applied (this study only made use of twelve) would improve the generalisability of these results. However, as the prevalence of certain onomatopoeic words in early childhood may also limit the number of useful references that can be drawn upon, the reliability of newly created onomatopoeic words, e.g. Hawk Talk's use of 'chwirk' and 'kee-eee-arrr' [10], for sound categorisation should also be considered. This is supported by our findings, which showed that participants were often surer about which descriptions were inappropriate, than were appropriate. Generating new onomatopoeic word sets (e.g. from card-sorting methods) may reverse this trend. Newly created onomatopoeic words may also be a means of bridging language differences and describing unfamiliar and complex sounds in nature.

This study's finding that some bird calls resulted in multiple high rated descriptions and lower levels of agreement, may also have resulted from differences in the aural qualities of the recordings used, i.e. some recordings may have been poorer than others leading to greater confusion regarding the appropriate descriptor. While it seems more likely that the supplied descriptions were in some cases inappropriate, further research could consider presenting multiple recordings of the same species' calls to improve internal validity.

This study made use of mTurk workers as participants, and recent research suggests differences in how these participants categorise audio in comparison to citizen scientists, at least when audio is accompanied by visualisations, and in terms of how many passes are required [4]. Thus, further work is needed to confirm the reliability of our current findings in non-mTurk populations. Further research could also consider how text descriptions subtract or add to the workload of annotators, and additionally, test this upon both crowdworkers and citizen scientists. However, little is known as to whether this study's participants varied in terms

of their interest for the subject matter, and it may be that crowdworkers and citizen scientists are not discrete groups in terms of their interests. Learning more about the motivations and interests of paid crowd workers may produce insights into how to best make use of their time and abilities in other ecological projects needing human intelligence. Relatedly, while our participants were 'Masters' (implying they had a good grasp of the English language) we do not know if English was their first language, and if not, what effect this may have had on their ratings. Future research on the use of onomatopoeia in sound categorisation tasks could consider contrasting participants who have the target language as a first or second language to learn if this is impactful.

The different methods that acoustic classification tasks make use of–visualisation, sound bites, onomatopoeia–could also be contrasted to learn if some methods offer complementary strengths and weaknesses. This in turn can inform the design of new audio annotation interfaces. Further research in this space could also validate these findings by seeing if agreement (regarding onomatopoeic descriptions of bird calls) can be reached using an expert sample. Relatedly, we cannot know for sure if some of our participants were perceiving the descriptions (e.g. 'whistle') as semantic constructs, and not onomatopoeia. While, the combination of both approaches in a single classification framework has been found to improve its flexibility and the accuracy of the labelling [with a range of human-made and natural sounds; 23], future research might usefully attempt to isolate the impact of each approach to commonly used bird sound descriptors.

## Conclusions

This research demonstrates that onomatopoeia is a valid method for non-experts to use in classifying some natural sounds. As such, it suggests a simple means of fostering greater citizen science engagement with bioacoustics science projects by showing that complex natural sounds might be intuitively summarised and understood. While the advantages of onomatopoeia used in concert with other methods of acoustic categorisation (visualisations, sound bites, machine learning) are yet to be explored, further exploration of its usefulness for citizen science projects is warranted, especially those contributing to the development of bioacoustics call library production. Our findings, showing multiple highly rated descriptions and patterns of high/low description ratings, also suggest citizen science acoustic annotation design solutions that cluster and contrast words, and additionally point to the potential of invented onomatopoeia. As such, this study adds onomatopoeia to the suite of methods that can be considered for use in acoustic categorisation tasks and provides directions for future research and design.

## Supporting information

**S1 File. All mean congruence ratings, descriptive statistics, and comparisons showing mean differences, p values and effect sizes.**
(XLSX)

**S2 File. Dataset containing raw scores of bird call description ratings, number of times audio was played, and confidence in ratings.**
(CSV)

**S1 Fig. All ten frequency charts.**
(PDF)

## Acknowledgments

We would like to thank Rebecca Dilworth for her work on the analysis and literature review, Sofia Woods who created the experimental architecture, and the participants for their time and input.

## Author Contributions

**Conceptualization:** Daniel Johnson, Paul Roe.

**Formal analysis:** Kellie Vella, Daniel Johnson.

**Funding acquisition:** Paul Roe.

**Investigation:** Daniel Johnson.

**Methodology:** Daniel Johnson, Paul Roe.

**Writing – original draft:** Kellie Vella.

**Writing – review & editing:** Kellie Vella, Daniel Johnson, Paul Roe.

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
