## [Decision Letter · Decision Letter 0]

2 Dec 2020

PONE-D-20-35160

Describing the sounds of nature: using onomatopoeia to classify bird calls for citizen science

PLOS ONE

Dear Dr. Vella,

Thank you for submitting your manuscript to PLOS ONE. After careful consideration, we feel that it has merit but does not fully meet PLOS ONE’s publication criteria as it currently stands. Therefore, we invite you to submit a revised version of the manuscript that addresses the points raised during the review process.

We look forward to receiving your revised manuscript.

Kind regards,

Almo Farina, PhD

Academic Editor

PLOS ONE

Journal Requirements:

Additional Editor Comments:

This paper has been considered of great interest by both reviewers, only minor changes are requested.

Reviewers' comments:

Reviewer's Responses to Questions

**Comments to the Author**

1. Is the manuscript technically sound, and do the data support the conclusions?

Reviewer #1: Partly

Reviewer #2: Yes

2. Has the statistical analysis been performed appropriately and rigorously? 

Reviewer #1: Yes

Reviewer #2: Yes

3. Have the authors made all data underlying the findings in their manuscript fully available?

Reviewer #1: Yes

Reviewer #2: Yes

4. Is the manuscript presented in an intelligible fashion and written in standard English?

Reviewer #1: Yes

Reviewer #2: Yes

5. Review Comments to the Author

Reviewer #1: This is a well written paper on an important topic of applying citizen science in classifying bird sounds. The structure of the paper and style of argumentation are well developed. The statistics is adequately used, as far as I am able to follow this. All the other aspects of paper (relevance for the field, used references, connection of data and results, etc) follow the standards of academic publishing.

I have some methodological concerns that I would like authors to address as minor revisions:

Lines 226-229. Please describe more in details, how was the list of vocal descriptors compiled. From what sources were descriptor words derived? Do the selected descriptors correspond to any functional classes of bird sounds (alarm calls, territorial calls, songs)? Where some candidates of descriptor words excluded from the list and if then on what basis? Where there any consideration of asking participants to name/describe the sounds in open ended question?

Lines 242-248. Was the background information on participants’ mother tongue and cultural/geographical location asked? Would authors consider cultural and linguistic issues relevant for the present study? As linguistic and cultural background of respondents presumably has an effect on what and how respondents perceive onomatopoetic words, please address this topic on discussion.

Lines 424-428. Please address in the Discussion the possible distinction between onomatopoeia and meanings (semantics) in regard to the analyzed descriptors. At present it is not clear whether your results reflect the measure of similarity between the descriptors and bird sounds (onomatopoeia) or correspondence between the semantic meaning of the descriptor concepts and the bird sounds. For instance, respondents probably have an idea/cognition what counts as a „whistle“, and could map bird sounds based on the correspondences in conceptual level. Please consider also to what extent this aspect should be taken into account in future research on the given topic.

Reviewer #2: The paper is of great interest to develop a new methodology especially in a prospective to analyzing big data.

Very few comments and suggestions:

Line 88: I don't understand "domain" in this context

Line 91: I suggest to include also ecoacoustics "bioacoustics and ecoacoustics research". In fact at line 95 we are referring to ecologists as well.

Line 130: the final sentence is a few obscure

Line 120-128: I suggest to reduce the text

Line 143: This sentence seems not necessary, see the successive section

Line 218: You are referring to ecoacoustics research but at line 44 you have used bio-acoustics. So please try to introduce better ecoacoustics before line 218.

6. PLOS authors have the option to publish the peer review history of their article (what does this mean?). If published, this will include your full peer review and any attached files.

Reviewer #1: No

Reviewer #2: No

---

## [Author Response · Author response to Decision Letter 0]

22 Dec 2020

We thank the reviewers for their feedback and have made the following changes in response.

Reviewer #1:

Lines 234-240: We have made additions to the Procedure describing how the list of vocal descriptors was compiled including their sources, types of calls, reason for exclusion, and why we preferred to provide a list of descriptors rather than letting participants describe them. We highlight that we were seeking broadly applicable descriptors that were commonly used amongst people with an interest in bird calls. 

Regarding the participants’ background information, we did not collect information regarding their first language or geographic location. Instead we now note that by including mTurk workers with a Masters rating, it was implied that they had a good grasp of English (added to line 261). However, lines 234-238 now acknowledge that we did not know if participants had English as a first language, and if they did not, how this may have impacted their ratings. As there is little research on this subject, we suggest the usefulness of future research contrasting participants who have the target language as a first or later language.

Regarding the distinction between semantics and onomatopoeia, we agree that we can’t know what someone’s conceptual construct of, for example, ‘whistle’ was. However, while this distinction is not the focus in the current study, this has been explored in other work (now described in lines 142-145). We also now discuss this distinction (lines 458-553) in relation to the provision of diverse types of information in audio categorisation tasks. Also, as noted in the Procedure, our choice of descriptors was due to seeking words commonly used to describe bird calls.

Reviewer #2: 

We appreciate the need for consistency in terminology but have chosen to remove mention of ecoacoustics throughout, as this study is focused specifically on bioacoustics. In addition, we remove mention of ‘ecologists’ (line 95) as it was not our intention to contrast ecologists and the general public (which is how it read), but instead the different requirements for call identification in sound bite sharing and when using long-duration recordings of the environment. We make changes to lines 94-96 that we hope clarify this distinction.

Line 88: ‘Domain’ has been replaced with ‘environmental conservation’.

Obscure or unnecessary text has been changed or removed at lines 120-128 and 148-149. 

We hope that these edits address the reviewers’ concerns.

---

## [Decision Letter · Decision Letter 1]

6 Apr 2021

Describing the sounds of nature: using onomatopoeia to classify bird calls for citizen science

PONE-D-20-35160R1

Dear Dr. Vella,

We’re pleased to inform you that your manuscript has been judged scientifically suitable for publication and will be formally accepted for publication once it meets all outstanding technical requirements.

Kind regards,

George Vousden

Division Editor

PLOS ONE

Additional Editor Comments (optional):

Reviewers' comments:

Reviewer's Responses to Questions

**Comments to the Author**

1. If the authors have adequately addressed your comments raised in a previous round of review and you feel that this manuscript is now acceptable for publication, you may indicate that here to bypass the “Comments to the Author” section, enter your conflict of interest statement in the “Confidential to Editor” section, and submit your "Accept" recommendation.

Reviewer #1: All comments have been addressed

Reviewer #2: All comments have been addressed

2. Is the manuscript technically sound, and do the data support the conclusions?

Reviewer #1: Yes

Reviewer #2: Yes

3. Has the statistical analysis been performed appropriately and rigorously? 

Reviewer #1: Yes

Reviewer #2: Yes

4. Have the authors made all data underlying the findings in their manuscript fully available?

Reviewer #1: Yes

Reviewer #2: Yes

5. Is the manuscript presented in an intelligible fashion and written in standard English?

Reviewer #1: Yes

Reviewer #2: Yes

6. Review Comments to the Author

Reviewer #1: (No Response)

Reviewer #2: Very well, the manuscript does not require further revisions all the comments and suggestions have been addressed

7. PLOS authors have the option to publish the peer review history of their article (what does this mean?). If published, this will include your full peer review and any attached files.

Reviewer #1: No

Reviewer #2: No

---

## [Editor Report · Acceptance letter]

8 Apr 2021

PONE-D-20-35160R1 

Describing the sounds of nature: using onomatopoeia to classify bird calls for citizen science 

Dear Dr. Vella:

I'm pleased to inform you that your manuscript has been deemed suitable for publication in PLOS ONE. Congratulations! Your manuscript is now with our production department. 

Kind regards, 

on behalf of

Dr. George Vousden 

Staff Editor

PLOS ONE